# A phylogenetic analysis of the wild *Tulipa* species (Liliaceae) of Kosovo based on plastid and nuclear DNA sequence

Avni Hajdari[1] | Bledar Pulaj[1] | Corinna Schmiderer[2] | Xhavit Mala[3] |
Brett Wilson[4] | Kimete Lluga-Rizani[1] | Behxhet Mustafa[1]

[1]Department of Biology, Faculty of
Mathematics and Natural Science, University
of Prishtina "Hasan Prishtina", Prishtinë,
Kosovo

[2]Institute of Animal Nutrition and Functional
Plant Compounds, University of Veterinary
Medicine, Vienna, Austria

[3]Sharri National Park, Prizren, Kosovo

[4]Department of Plant Sciences, University of
Cambridge, Cambridge, UK

**Correspondence**
Avni Hajdari, Department of Biology, Faculty
of Mathematics and Natural Science,
University of Prishtina "Hasan Prishtina",
Mother Theresa St. 10000 Prishtinë, Kosovo.
Email: avni.hajdari@uni-pr.edu

**Abstract**

In Kosovo, the genus *Tulipa* is represented by eight taxa, most of which form a species complex surrounding *Tulipa scardica*. To investigate the phylogenetic relationship of these *Tulipa* species a Bayesian analysis was undertaken using the ITS nuclear marker and *trnL-trnF*, *rbcL* and *psbA-trnH* plastid markers. The resulting phylogenetic trees show that Kosovarian *Tulipa* species consistently group into two main clades, the subgenera *Eriostemones* and *Tulipa*. Furthermore, our analyses provide some evidence that the subspecies of *Tulipa sylvestris* are genetically distinguishable, however not significantly enough to support their reclassification as species. In contrast, the markers provide some novel information to reassess the species concepts of the *T. scardica* complex. Our data provide support for the synonymisation of *Tulipa luanica* and *Tulipa kosovarica* under the species *Tulipa serbica*. Resolution and sampling limitations hinder any concrete conclusion about whether *Tulipa albanica* and *T. scardica* are true species, yet our data do provide some support that these are unique taxa and therefore should continue to be treated as such until further clarification. Overall, our work shows that genetic data will be important in determining species concepts in this genus, however, even with a molecular perspective pulling apart closely related taxa can be extremely challenging.

**KEYWORDS**
Balkans, barcoding, ITS, phylogenetics, species concepts I *trnL-trnF*, Tulipa

## 1 | INTRODUCTION

Species of the genus *Tulipa* L. (Liliaceae) have great economic, horticultural, and ecological value[1] while also being culturally significant in many areas of the world.[2] They are bulbous monocots characterized by a diverse range of variable vegetative and floral traits, which were traditionally used to define species concepts in this genus. Furthermore, the vegetative and floral traits often show a high degree of plasticity, sometimes, even within populations of a species.[2-4] Due to this and the long horticultural history of tulips, creating a stable taxonomic framework for the genus has been extremely difficult, despite the existence of a large body of literature,[3-6] and so classifications of *Tulipa* have been revised several times,[7] The total number of extant *Tulipa* species varies between publications, although generally ranges from 40 to 150 species.[5,8] In the *World Checklist of Selected Plant Families*,[9] 516 names are listed for *Tulipa*, but only 102 taxa have been accepted, while in the *Plant List*[10] 499 names are listed for *Tulipa* and 120 taxa have been accepted. According to the most complete evaluation of the genus to date,[2] only 76 species are accepted, but since this work, a number of new species have been described.[11-14] The

number of *Tulipa* species native to the Balkan Peninsula is only a small proportion of the global diversity, varying from 15[15] to 22[9] species. In Kosovo, the genus *Tulipa* is represented by eight taxa (six species and two subspecies), belonging to the subgenera *Eriostemones* and *Tulipa*. In general, researchers working on these species have used different morphological traits to define the taxonomic relationship between them. The subgenus *Eriostemones*, is generally represented by *Tulipa sylvestris* and at the lower taxonomic level by two subspecies,[16] *T. sylvestris* subsp. *sylvestris* only accepted by the *World Checklist of Selected Plant Families*[9] and *Tulipa sylvestris* subsp. *australis* (Link) Pamp (accepted subsp.). While the subgenus *Tulipa* is represented by several species, *Tulipa gesneriana* L.[16] is sometimes treated as a wild species, although it is not thought to grow in a truly wild state[2] and is believed to be a complex hybrid derived from *T. agenensis* DC, *T. armena* Boiss, *Tulipa suaveolens* Roth and so on.[9] It is here not treated as a true species in line with previous research, but is included in a range of analyses.[2] *Tulipa scardica* Bornm. which has a distribution that encompasses Southern Kosovo and North Macedonia.[17] In Kosovo, it occurs near the village Krivenik, close to the border of North Macedonia. It is synonymised as *T. gesneriana* L.,[2,6,10] accepted as a species by the *World Checklist of Selected Plant Families*,[9] but not accepted by Flora Europea.[18] *Tulipa serbica* Tatic & Krivošej occurs on serpentine soil in the South of Serbia (community Knjaževac: Mt. Rogozna near Donja Kamenica) and Northern Kosovo (Beli Laz hill, near Ibar river).[19] *Tulipa kosovarica* Kit Tan, Shuka & Krasniqi is endemic to Kosovo, in the serpentine area of Mirusha region at the foot of Mt. Koznik, between Mrasori and Llapçevë villages,[20] as well as in the localities Guriç, Llapushnik, Qafë Prush and Devë.[16] *Tulipa luanica* Millaku is also endemic to Kosovo found growing on limestone substrate on Mt. Pashtrik, located in the district of Prizren, Southern Kosovo, near the border with Albania.[11] *Tulipa albanica* Kit Tan & Shuka was originally described as a new species from a locality in Albania[21] (Kukësi district: from Kolshi to Surroj village, on serpentine slopes), but has been recently found growing in the Kosovar village of Deva.[16] *T. scardica*, *T. serbica*, *T. albanica*, *T. kosovarica*, and *T. luanica* are all morphologically similar[19-21] and form the species complex known as the *T. scardica* complex (*scardica* complex), named after the oldest species in the group.[2] Due to the similarities between these species, they have sometimes been treated as synonyms, and are often erroneously identified and misclassified.

Studies focused on defining species concepts within the *scardica* complex have primarily used morphological characteristics and geographical distributions. However, in addition, karyological analyses have been undertaken for *T. albanica*,[21] and *T. luanica*,[11] as well as measurements of nuclear genome size (DNA 2C-values) for *T. albanica*,[21,22] *T. scardica*,[6] *T. kosovarica*, and *T. luanica*.[22] However, DNA content and cytogenetic analyses have not been undertaken for all species present in Kosovo and so understanding of species relationships is currently limited.

DNA/molecular markers have emerged in the last few decades as a powerful tool in plant systematics and have become an important, inexpensive, reliable technique for exploring phylogenetic relationships.[23] Molecular phylogenetic analysis using sequences from nuclear ribosomal DNA (nrDNA) and chloroplast DNA (cpDNA) have previously been successfully used to determine relationships between species within the genus *Tulipa*. Thus, we decided to use *Tulipa* DNA sequences from the ITS region,[2,7,24,25] *trnL-trnF* region,[26] *psbA-trnH* region, and *rbcL* region[27] to undertake a phylogenetic analysis of Kosovarian tulip diversity. This work aimed to improve understanding of species concepts across the wild-growing *Tulipa* species of Kosovo, especially the *scardica* complex, with a view to inform tulip conservation, evolutionary understanding, and the broader taxonomic positioning of Kosovarian tulip species.

## 2 | RESULTS

The ITS sequences (ITS1, complete 5.8S rDNA gene, ITS2 and a small part of 26S rDNA gene) of *Tulipa* species in the dataset ranged from 616 to 655 bp. The in-group alignment included 66 ambiguous positions. Sixty-seven positions were potentially informative, 33 potentially informative indels, and 60.0% G + C content (Table 1). The sequence length of ITS1 ranged between 229 and 233 bp, 5.8S rDNA between 162 and 166 bp, ITS2 between 225 and 231 bp and 26S rDNA (partial) was consistently 26 bp. Tulip samples showed an average of 141 and 143 conserved sites for ITS1 and ITS2, respectively. The *trnL-trnF* sequences of *Tulipa* species in the dataset ranged from 765 to 788 bp in length. The in-group alignment had 46 ambiguous positions. Analyzed sequences showed eight potentially informative characters, 16 potentially informative indels, and 31.2% G + C content (Table 1). The *trnL-trnF* region was made up of *trnL* 631 to 692 bp, *trnF* 57-64 bp and IGS 25 bp for each sequence, respectively. The *rbcL* sequence length in the dataset ranged from 488 to 597 bp. In-group alignment includes three ambiguous positions. Analyzed sequences showed three potentially informative characters, five potentially informative indels, and 44.0% G + C content (Table 1). The *psbA-trnH* sequences in the dataset ranged from 488 to 597 bp in length. The in-group alignment had 35 ambiguous positions. Analyzed sequences showed 15 potentially informative characters, 93 potentially informative indels, and 32.6% G + C content (Table 2). The combined ITS + *trnL-trnF* + *psbA-trnH* + *rbcL* sequences for species ranged from 2405 to 2469 bp in length. The alignment showed 134 ambiguous positions, 2272 conserved sites, 113 potentially informative characters, and 125 potentially informative indels, and an average 42.1% G + C content (Table 1).

### 2.1 | Phylogenetic analysis

In total, 106 sequences were used in the phylogenetic analysis. The phylogenetic trees for all datasets (separate ITS, *trnL-trnF*, *psbA-trnH*, and *rbcL* trees as well as the combined ITS + *trnL-trnF* + *psbA-trnH* + *rbcL* datasets) were generated through a Bayesian analyses. Resolution was relatively weak for all trees produced from single markers while the best resolution was obtained from the phylogenetic tree created using the combined ITS + *trnL-trnF* + *psbA-trnH* + *rbcL* dataset.

**TABLE 1** Data set and parsimony-based tree characteristics for ITS and *trnL-trnF*, *rbcL*, and *psbA-trnH* analyses

| Parameters | ITS | *trnL-trnF* | *rbcl* | *psba-trnH* | Combined *trnL-trnF* + *rbcL* + *psbA* |
|---|---|---|---|---|---|
| No. of taxa | 16 | 14 | 13 | 11 | 10 |
| No. of sequences | 31 | 28 | 29 | 26 | 21 |
| Alignment length (bp) | 657 | 817 | 597 | 486 | 2551 |
| Sequence minimum length (bp) | 616 | 765 | 488 | 413 | 2405 |
| Sequence maximum length (bp) | 655 | 788 | 597 | 469 | 2469 |
| Number of ambiguous positions: ingroup | 66 | 46 | 3 | 35 | 134 |
| Number of ambiguous positions: outgroup | 77 | 60 | 8 | 153 | 289 |
| Conserved characters | 555 | 762 | 586 | 357 | 2272 |
| Variable characters | 100 | 33 | 11 | 111 | 242 |
| Potentially informative characters | 67 | 17 | 6 | 15 | 113 |
| Number of potentially informative indels | 33 | 16 | 5 | 93 | 125 |
| G + C contents | 60.07 | 31.17 | 43.90 | 32.56 | 42.08 |

## 2.2 | ITS region

The phylogenetic analysis based on 31 ITS sequences is shown in Figure 1. The generated tree shows that the *Tulipa* taxa are divided into two main clades with strong support (BPP = 1). The first clade includes specimens of the subgenus *Eriostemones* (*T. sylvestris*, including both subspecies), while the second clade includes members of the subgenus *Tulipa* (*T. albanica*, *T. kosovarica*, *T. luanica*, *T. scardica*, *T. serbica*, *Tulipa ulophylla*, *Tulipa tschimganica*, *T. suaveolens*, *Tulipa julia*, and *T. gesneriana*). In the first clade, the wild-collected specimens of *T. sylvestris* subsp. *sylvestris* (T21 and T23) are separated from the wild *T. sylvestris* subsp. *australis* (T22), while all wild-collected specimens are more closely related to each other than to the *T. sylvestris* subsp. *sylvestris* sequence obtained from GenBank (BPP 1%). In the second clade, all species from the *scardica* complex form a single clade (*T. albanica*, *T. kosovarica*, *T. luanica*, *T. scardica*, and *T. serbica*), with specimens of *T. ulophylla*, *T. x tschimganica*, *T. suaveolens*, *T. julia*, and *T. gesneriana* all more distantly related. The species *T. x tschimganica* (section *Spiranthera*), *T. ulophylla* (section *Tulipanum*), and *T. julia* were all identifiable as separate taxonomic entities (BPP = >0.9), while *T. suaveolens* and *T. gesneriana* formed a strongly supported clade (BPP = 1) that was sister to the *scardica* complex, indicating that the sequences under the name *T. gesneriana* may in fact be *T. suaveolens*.

## 2.3 | trnL-trnF region

The phylogenetic tree obtained from 28 *trnL-trnF* sequences was again divided into two major clades representing the two subgenera sampled (Figure 2). The first clade, *Eriostemones* (*T. sylvestris* including both subspecies), was strongly supported by Bayesian analyses (BPP = 1), although the structure of the tree slightly differed from the phylogenetic tree created using ITS data; the *T. sylvestris* subsp. *australis* specimen (T22) was more closely related to a *T. sylvestris* subsp. *sylvestris* specimen (T23) than both wild-collected *T. sylvestris* subsp.

*sylvestris* to each other (T21 and T23). The structure of the second clade, *Tulipa*, varied somewhat more from that of the ITS region. The specimens of this subgenus form two separate subclades; the first strongly supported subclade (BPP = 1) includes *T. kosovarica*, *T. luanica*, and *T. serbica*, albeit these species concepts are not monophyletic. The second subclade consists of *T. albanica*, *T. scardica*, *T. julia*, *T. ulophylla*, *T. suaveolens*, and *T. x tschimganica*, which had strong support for the grouping (BPP 1) but lacked any discernible structure within.

## 2.4 | psbA-trnH region

Twenty-six *psbA-trnH* sequences were used to construct a phylogenetic tree, which again resulted in the clear division of the sequences into two major clades (Figure 3). Again, the first strongly supported clade (BPP = 1) consists of members of the subgenus *Eriostemones*, with, similarly to the tree generated based on ITS sequences, wild *T. sylvestris* subsp. *sylvestris* separated from *T. sylvestris* subsp. *australis*. Surprisingly, the outgroup specimen *A. erythronioides* fell within the *Eriostemones* clade suggesting this may be a poor marker for taxonomic understanding. The second strongly supported clade (BPP = 1) consists of members of the subgenus *Tulipa*. The specimens of subgenus *Tulipa* appear to divided into two groups; the first consisting of all *T. albanica*, specimens, while the second group encompasses specimens of *T. scardica*, *T. kosovarica*, *T. luanica*, and *T. serbica*, although none of these were monophyletic with the tree lacking structure (BPP = <0.5).

## 2.5 | rbcL region

The phylogenetic tree obtained from 29 *rbcL* sequences was the least informative single marker (Figure 4) with generally low posterior probability scores. Even so, the marker was able to distinguish between

**TABLE 2** Basic characteristics of the collection sites, voucher information, and GenBank (https://www.ncbi.nlm.nih.gov/nuccore/?term=tulipa+kosovo) accession numbers of the *Tulipa* samples used for this study

| Potential species | Sequence_ID | Collection locality | Country | Longitude | Latitude | Altitude | ITS accession number | trnL-trnF accession number | rbcl accession number | psba-trnH accession number | Uni. of Prishtina herb. Acces. no. |
|---|---|---|---|---|---|---|---|---|---|---|---|
| T. albanica | T._albanica_T1 (yellow flower) | Surroj | Albania | 42° 2.744'N | 20° 20.037'E | 622 | MN336199 | MN446897 | MZ147066 | MZ147043 | 00000158 |
| T. albanica | T._albanica_T2 (reddish maroon flower) | Surroj | Albania | 42° 2.744'N | 20° 20.037'E | 622 | MN336200 | MN446898 | MZ147067 | MZ147044 | 00000157 |
| T. albanica | T._albanica_T3 (reddish maroon /yellow flower) | Surroj | Albania | 42° 2.744'N | 20° 20.037'E | 622 | MN336201 | MN446899 | MZ147068 | MZ147045 | 00000156 |
| T. kosovarica | T._kosovarica_T4 | Goriç | Kosovo | 42° 26.689'N | 20° 45.337'E | 659 | MN336202 | MN446900 | MZ147069 | MZ147046 | 00000155 |
| T. kosovarica | T._kosovarica_T5 | Goriç | Kosovo | 42° 26.689'N | 20° 45.337'E | 659 | MN336203 | MN446901 | MZ147070 | MZ147047 | 00000154 |
| T. kosovarica | T._kosovarica_T6 | Koznik | Kosovo | 42° 30.334'N | 20° 33.987'E | 425 | MN336204 | MN446902 | MZ147071 | MZ147048 | 00000153 |
| T. kosovarica | T._kosovarica_T7 | Koznik | Kosovo | 42° 30.334'N | 20° 33.987'E | 425 | MN336205 | MN446903 | MZ147072 | MZ147049 | 00000152 |
| T. kosovarica | T._kosovarica_T8 | Koznik | Kosovo | 42° 30.334'N | 20° 33.987'E | 425 | – | MN446904 | MZ147073 | MZ147050 | 00000151 |
| T. species | T._species_T9 | Krojmir | Kosovo | – | – | – | MN336206 | – | MZ147074 | MZ147051 | 00000150 |
| T. luanica | T._luanica_T10 | Pashtrik | Kosovo | 42° 14.966'N | 20° 30.399'E | 1041 | – | MN446905 | MZ147075 | MZ147052 | 00000149 |
| T. luanica | T._luanica_T11 | Pashtrik | Kosovo | 42° 14.966'N | 20° 30.399'E | 1041 | MN336207 | MN446906 | MZ147076 | MZ147053 | 00000146 |
| T. luanica | T._luanica_T12 | Pashtrik | Kosovo | 42° 14.966'N | 20° 30.399'E | 1041 | MN336208 | MN446907 | MZ147077 | MZ147054 | 00000147 |
| T. luanica | T._luanica_T13 | Qafë Prush | Kosovo | 42° 18.275'N | 20° 23.529'E | 580 | MN336209 | MN446908 | MZ147078 | MZ147055 | 00000148 |
| T. luanica | T._luanica_T14 | Qafë Prush | Kosovo | 42° 18.275'N | 20° 23.529'E | 580 | MN336210 | MN446909 | MZ147079 | MZ147056 | 00000145 |
| T. scardica | T._scardica_T15 | Krivenik | Kosovo | 42° 6.254'N | 21° 14.958'E | 575 | MN336211 | MN446910 | MZ147080 | MZ147057 | 00000167 |
| T. scardica | T._scardica_T16 | Krivenik | Kosovo | 42° 6.254'N | 21° 14.958'E | 575 | MN336212 | – | MZ147081 | MZ147058 | 00000166 |
| T. scardica | T._scardica_T17 | Krivenik | Kosovo | 42° 6.254'N | 21° 14.958'E | 575 | MN336213 | – | MZ147082 | MZ147059 | 00000165 |
| T. serbica | T._serbica_T18 | Serboc | Kosovo | 42° 58.067'N | 20° 49.757'E | 596 | MN336214 | MN446911 | MZ147083 | MZ147060 | 00000164 |
| T. serbica | T._serbica_T19 | Serboc | Kosovo | 42° 58.067'N | 20° 49.757'E | 596 | MN336215 | MN446912 | MZ147084 | MZ147061 | 00000163 |
| T. serbica | T._serbica_T20 | Serboc | Kosovo | 42° 58.067'N | 20° 49.757'E | 596 | MN336216 | MN446913 | MZ147085 | MZ147062 | 00000162 |
| T. sylvestris ssp. sylvestris | T._sylvestris_ssp._sylvestris_T21 | Goriç | Kosovo | 42° 26.747'N | 20° 45.293'E | 665 | MN336217 | MN446914 | MZ147086 | MZ147063 | 00000161 |
| T. sylvestris ssp. australis | T._sylvestris_ssp._australis_T22 | Devë | Kosovo | 42° 19.950'N | 20° 20.517'E | 700 | MN336218 | MN446915 | MZ147087 | MZ147064 | 00000160 |
| T. sylvestris ssp. sylvestris | T._sylvestris_ssp._sylvestris_T23 | Devë | Kosovo | 42° 19.950'N | 20° 20.517'E | 700 | MN336219 | MN446916 | MZ147088 | MZ147065 | 00000159 |

**TABLE 2** (Continued)

| Potential species | Sequence_ID | Collection locality | Country | Longitude | Latitude | Altitude | ITS accession number | trnL-trnF accession number | rbcl accession number | psba-trnH accession number | Uni. of Prishtina herb. Acces. no. |
|---|---|---|---|---|---|---|---|---|---|---|---|
| T. ulophylla | T._ulophylla_T24 | – | – | – | – | – | HF952978 | HF953003 | – | – | – |
| T. tschimganica | T._tschimganica_T25 | – | – | – | – | – | HF952976 | HF953001 | KM085539 | – | – |
| T. sylvestris ssp. sylvestris | T._sylvestris_ssp._sylvestris_T26 | – | – | – | – | – | HF952974 | HF952999 | KM085538 | – | – |
| T. suaveolens | T._suaveolens_T27 | – | – | – | – | – | MK33446 | HF952998 | – | – | – |
| T. julia | T._julia_T28 | – | – | – | – | – | HF952964 | HF952989 | – | – | – |
| T. gesneriana | T._gesneriana_T29 | – | – | – | – | – | MK335217 | – | KP711981 | – | – |
| T. gesneriana | T._gesneriana_T30 | – | – | – | – | – | MK335224 | – | – | – | – |
| Amana edulis | A._edulis_T31 | – | – | – | – | – | MN173164 | HF953006 | KC796897 | NC034707 | – |
| Amana erythronioides | A._erythronioides_T32 | – | – | – | – | – | HF952982 | HF953007 | NC03463 | EU939293 | – |
| Erythronium japonicum | E._japonicum_T33 | – | – | – | – | – | EU912083 | HF953009 | D28156 | EU939295 | – |

the members of the subgenus *Eriostemones* and that of the subgenus *Tulipa* albeit with very limited resolution. In the *Eriostemones* clade, the specimens of *T. sylvestris* subsp. *sylvestris* formed a clade separate from *T. sylvestris* subsp. *australis* (BPP 0.9) Within the *Tulipa* clade the Bayesian analysis provided extremely limited resolution (BPP < 0.5) to distinguish between taxa especially those of the *scardica* complex (*T. albanica*, *T. kosovarica*, *T. luanica*, *T. scardica*, and *T. serbica*). Nonetheless, there was some support that *T. x tschimganica* and *T. gesneriana* were genetically distinct from the specimens in the *scardica* complex (BPP = 0.95). Here, we also note that *Erythronium japonicum* appears more closely related to *Tulipa* specimens than to *Amana* specimens, which contradicts the expected relationship of these genera providing some evidence that this is not a taxonomically informative marker.

## 2.6 | Combined ITS, *trnL-trnF*, *psbA-trnH*, and *rbcL* dataset

The phylogenetic tree obtained from the combined ITS, *trnL-trnF*, *psbA-trnH*, and *rbcL* sequences provided the most strongly supported tree structure for the specimens analyzed (Figure 5). The phylogenetic tree is divided into two main clades, the subgenus *Eriostemones* and the subgenus *Tulipa* with strong support for this separation (BPP = 1). Within the *Eriostemones* subgenus, the specimens of *T. sylvestris* subsp. *sylvestris* fall together with *T. sylvestris* subsp. *australis* sister to these (BPP = 1). Within the *Tulipa* clade, the analyzed taxa divided into three clear genetically distinct clades of the tree. Both *T. albanica* and *T. scardica*, appear as taxonomically distinct clades, although *T. scardica* is only represented by a single specimen (BPP = 1), while a group consisting of *T. kosovarica*, *T. luanica*, and *T. serbica* created a third separate clades of the tree, which was strongly supported (BPP = 1). Within this last grouping the species concepts are not monophyletic. We also note here that within the outgroup *Amana edulis* and *A. erythronioides* do not fall together as expected; however, all members of the outgroup do sit outside the *Tulipa* clade.

## 3 | DISCUSSION

In this study, we use the genetic markers ITS, *trnL-trnF*, *psbA-trnH*, and *rbcL* to undertake a molecular phylogenetic analysis of Kosovarian tulip diversity. Our data highlight the informativeness and limitations of the ITS nuclear marker[2,7,24,25] and plastid markers *trnL-trnF*,[26] *rbcL*,[28-30] and *psbA-trnH*[30,31] in investigating evolutionary relationships between species of wild *Tulipa*. In general, we found that subgenera can be reliable separated by a range of single genetic markers; however, that separating more closely related species requires a combination of markers. Our most informative tree provides evidence that the *scardica* complex has been over split and specifically that *T. luanica* and *T. kosovarica* should be synonymised under *T. serbica*. While our data also provide some support for the existence of *T. albanica* and *T. scardica* as unique taxa, as well providing some evidence that the

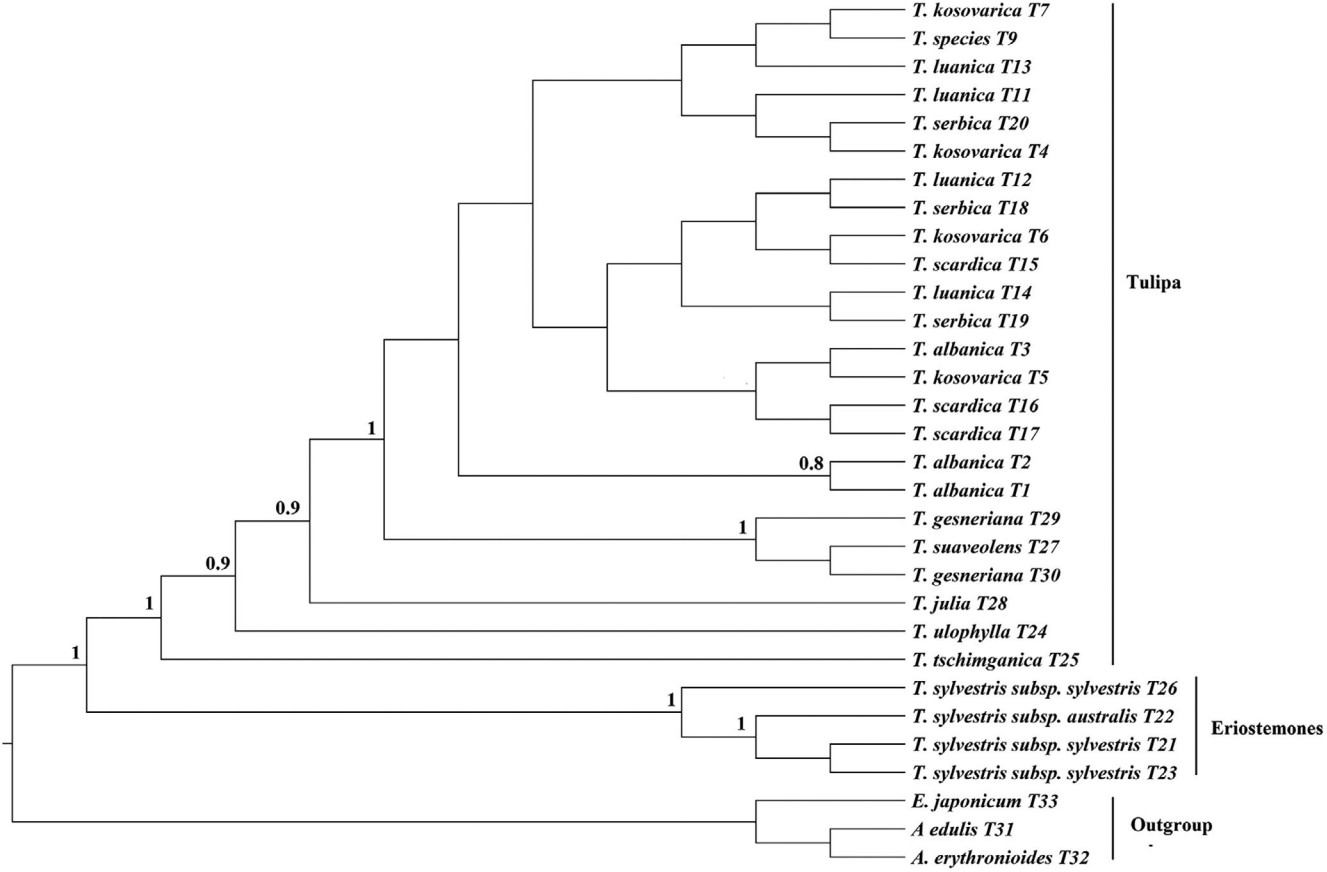

**FIGURE 1** Phylogenetic trees based on ITS sequences, including posterior probabilities (BPPs) (>0.5) provided above each node

subspecies of *T. sylvestris* can be distinguished genetically although should be maintained as a single species.

In general, phylogenetic trees generated using ITS sequence data had better resolution than those generated from single plastid markers, including the *trnL-trnF* marker which is in line with previous research.[7,32] The *rbcL* tree was the least informative as it had extremely weak resolution across the analyzed taxa, which supports previous reports of the marker performing poorly.[28-30] The *psbA-trnH* marker provided somewhat better resolution than *rbcL*,[30,31] but still lacked enough informative sites to separate the *scardica* complex and also unexpectedly placed an *Amana* specimen within the *Tulipa* clade showing it is not necessarily a reliable genetic marker. Our phylogenetic analyses also showed that the unidentified *Tulipa* species (sample T9, Table 2) sequenced from herbarium material at the Herbarium of the University Prishtina, falls into the *scardica* complex, but we lack the resolution to identify it as an existing or new species. It is, therefore, clear from our work and previous research that single genetic markers can only provide reliable resolution at the subgenera level.[2]

The use of sections within the genus *Tulipa* was actively discouraged[2] until further in-depth genetic studies could be undertaken. Yet, we wanted to briefly explore how our ITS tree fits into the taxonomic framework developed by Zonneveld.[6] The phylogenetic tree based on the ITS marker we generated had monophyletic groups that represented the *Eriostemones* section *Sylvestres* and the *Tulipa* sections

*Spiranthera* and *Tulipa*. Yet, our tree shows that the section *Tulipanum* in the *Tulipa* subgenera does not form a monophyletic group, with the specimen of *T. julia* shown to be more closely related to the species of the section *Tulipa* than to *T. ulophylla* of the same section. There are significant limitations in our assessment of sections of the genus *Tulipa* both in terms of the genetic marker used as well as in the extremely poor species representation. We therefore do not make any conclusive statements about the use of sections in the genus *Tulipa* but do note that these may not all hold as more genetic data become available.

Unsurprisingly, our most informative tree was generated using the combined dataset that included all the markers (ITS, *trnL-trnF*, *psbA-trnH*, and *rbcL*). This, like the single marker trees, separated the *Tulipa* taxa into two main clades, representing the subgenus *Eriostemones* and the subgenus *Tulipa*, which are clearly stable monophyletic taxonomic groupings.[2,7] Among the newly sequenced species of the *Eriostemones* clade, there was some distinguishable difference between *T. sylvestris* subsp. *australis* (Link) Pamp and *T. sylvestris* subsp. *sylvestris* from Kosovo. Our work therefore suggests that these subspecies should continue to be treated as separate taxa; however, within our work, we did not incorporate enough specimens or have the resolution to classify these as unique species. These subspecies are known to have different chromosome numbers, with *T. sylvestris* subsp. *australis* a diploid form of *T. sylvestris*, and *T. sylvestris* subsp.

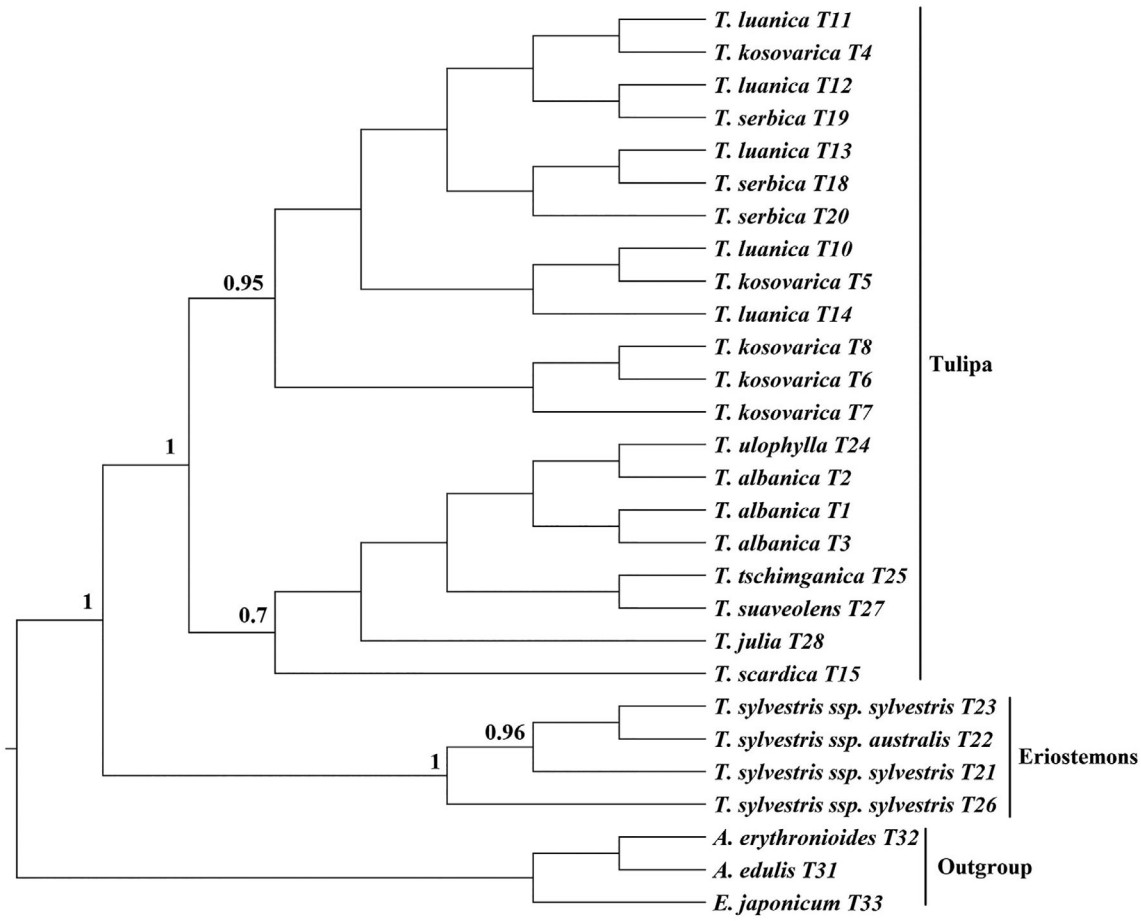

**FIGURE 2** Phylogenetic trees based on *trnL-trnF* sequences, including posterior probabilities (BPPs) (>0.5) provided above each node

*sylvestris* encompassing triploid or tetraploid forms of *T. sylvestris*.[5] Yet, the native range of these subspecies remains unclear, and many morphologically intermediate forms are known to occur in the wild.[2] Further cytotaxonomic studies will therefore be needed to investigate the chromosome numbers of the specimens located in Kosovo to confirm their taxonomic identity, while extensive in-depth molecular work will be needed to unentangle this widespread, complicated taxon. In the subgenus *Tulipa*, the grouping together of the species *T. scardica*, *T. serbica*, *T. albanica*, *T. kosovarica*, and *T. luanica* into a clade provides strong evidence of a close relationship between these taxa, confirming the existence of the *scardica* complex.[2] Our combined tree highlighted the genetic distinctness of *T. albanica* and *T. scardica* from the other species in this complex, while leaving the other three taxa in a clade where none were monophyletic. This provides evidence for the over splitting of this complex and the need to synonymize some of the taxa under one species name, specifically *T. luanica* and *T. kosovarica* under *T. serbica*.

The *scardica* complex remains a controversial group of species due to the many morphological similarities between these taxa. There has been significant confusion around species concepts, including in the use of the name *T. gesneriana*. In some instances, *T. scardica* has been synonymized under the name *T. gesneriana*, however, *T. gesneriana* is likely not a true species.[2] This taxonomic confusion is highlighted again in the varied acceptance of *T. gesneriana* as a species across different classification bodies; it is not accepted by Flora Europea,[18] but is by the *World Checklist of Selected Plant Families*.[9] Today, there are five species recognized as part of this complex.

*T. scardica* was the first species described from this complex,[33] and individuals of this species show significant variation in several morphological characters, such as leaf form, flower color, length of filaments, and anthers in different distribution areas.[17]

*Tulipa serbica*, the second species named in this complex was described from Mt Rogozna[34] and was originally thought to be a population of *T. scardica*, before being described as a new species.[20] Both species are thought to be closely related with *T. serbica* only morphologically differing from *T. scardica* in its paler, unspotted periapt segments, pale (not blackish) staminal filaments, dull violet (not yellowish), and acute anthers.[19]

*T. albanica* was recorded as a new species in Northeast Albania in 2010; it has recently been found growing in Kosovo as well.[16] It shows significant variation in several morphological characters from the other species in the group; it has a unique combination of yellow perianth bases without black blotches, yellow filaments, and violet-purple pollen.[11,20] The plant's campanulate flowers exist in two color forms, yellow to golden-yellow or carmine-scarlet turning deep reddish maroon, with a dominance of the golden-yellow flowers.[21] Some

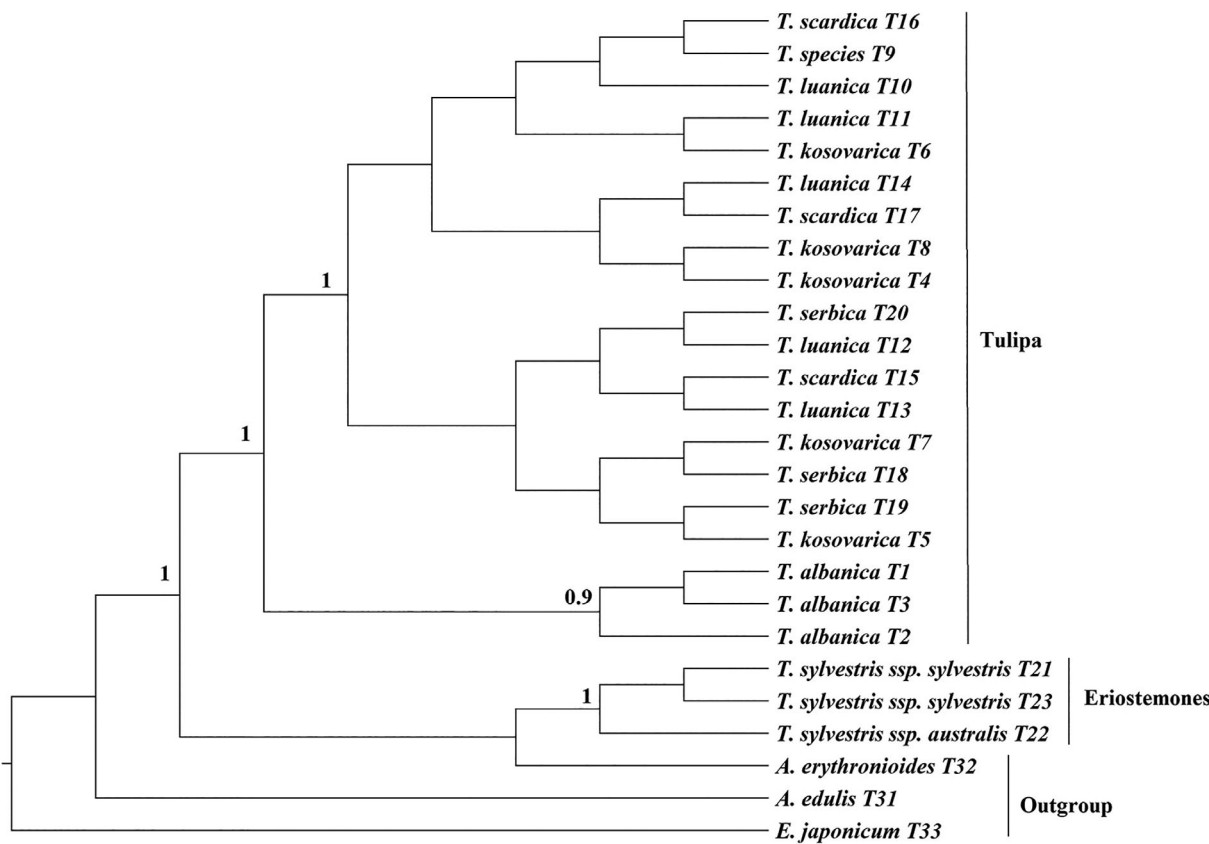

**FIGURE 3**    Phylogenetic trees based on *psbA-trnH* sequences, including posterior probabilities (BPPs) (>0.5) provided above each node

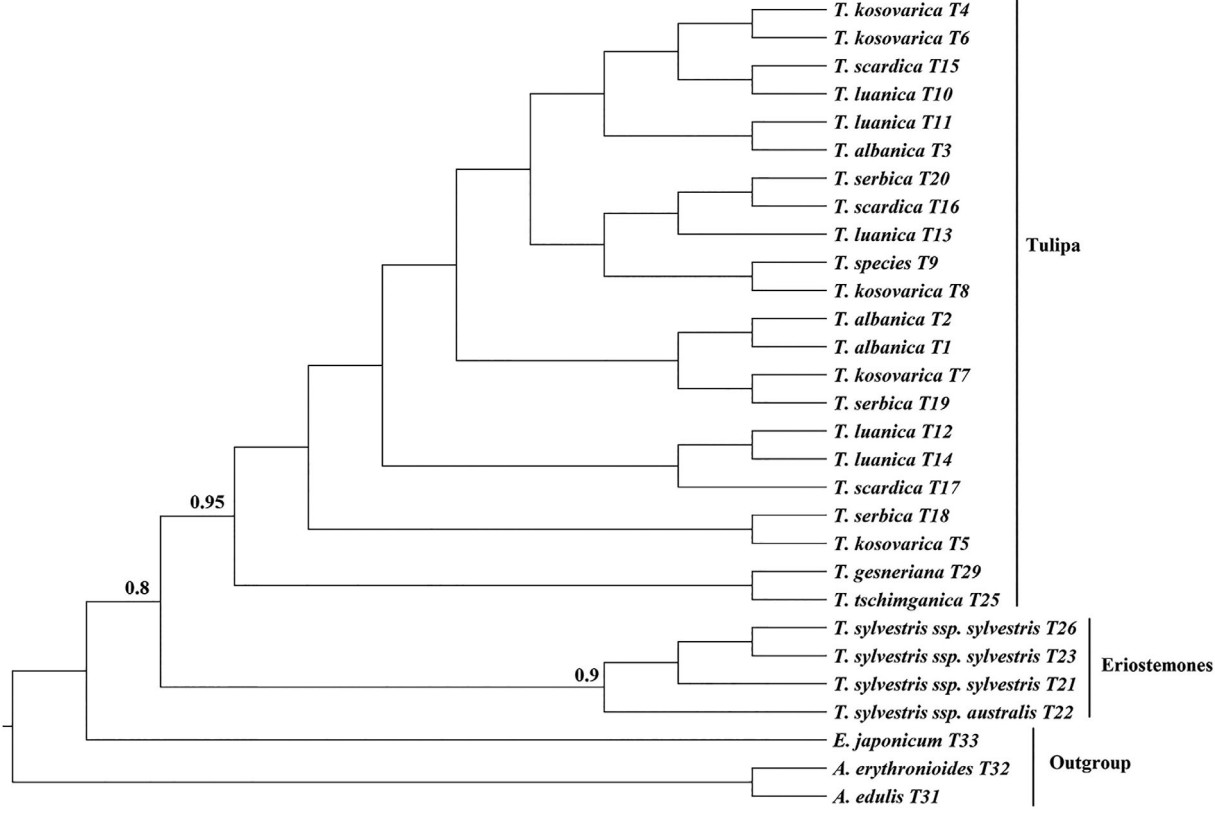

**FIGURE 4**    Phylogenetic trees based on *rbcL* sequences, including posterior probabilities (BPPs) (>0.5) provided above each node

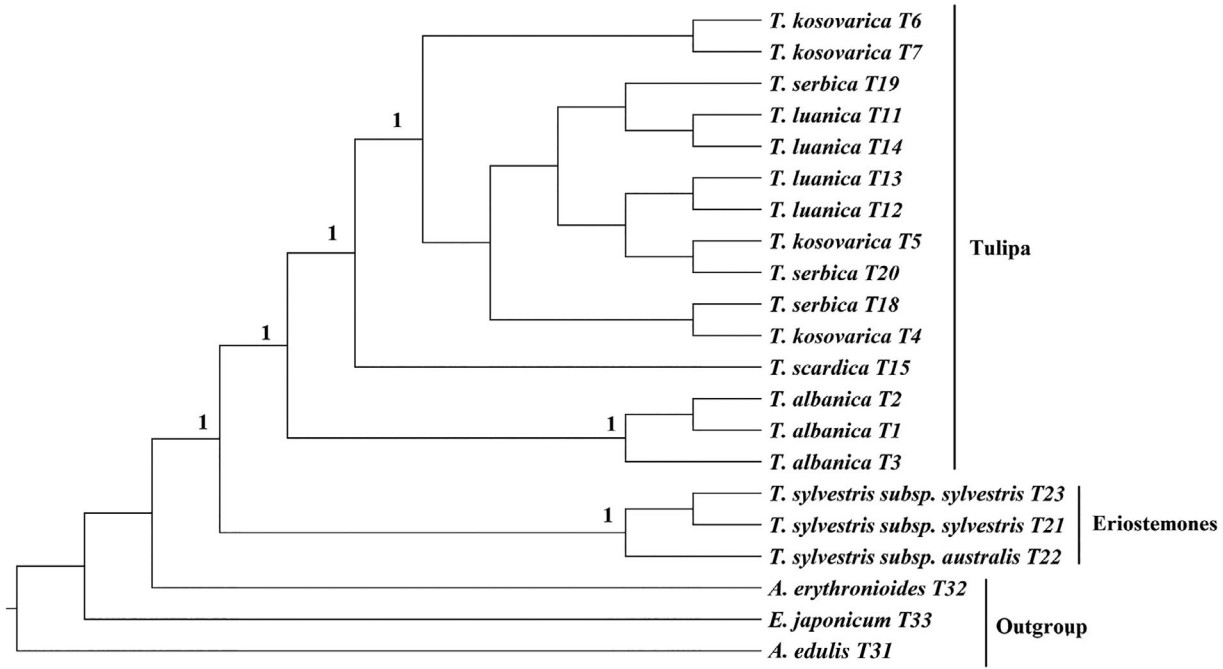

**FIGURE 5** Phylogenetic trees based on a combined ITS+*trnL-trnF*+ *rbcL* + *psbA-trnH* sequence set including, including posterior probabilities (BPPs) (>0.5) provided above each node

individuals have an intermediate color of yellow to reddish maroon. Yet, *T. albanica* also shares many morphological similarities with *T. scardica*, *T. serbica*, *T. kosovarica*, and *T. luanica*.

*T. kosovarica* collected for the first time along the Mrasori river (Mirusha region) at the foot of Mt Kozniku in 2010 was again originally thought to be a population of *T. scardica*,[21] but in 2012, the material was revisited and described as a new species.[20] Later, this species was recorded from several other locations such as Guriç, Llapushnik, Qafë Prush and Devë.[16] *T. kosovarica* differs from *T. scardica* due to its white or whitish perianth base that is sometimes masked by obtrullate patches of maroon and violet, while *T. albanica* differs from this species by having yellow perianth bases without black blotches.[20]

*T. luanica* is the most recent species described as a member of the *T. scardica* complex[11] that shares many morphological characters with *T. albanica*, *T. kosovarica*, and *T. serbica*. However, *T. luanica* also differs in several characters, including that it exclusively grows on limestone substrate rather than the serpentine substrate which other species grow on.

Across the literature, flower color has been one of the main characters used to discriminate the species of the *scardica* complex, but there is considerable variation in flower color within species.[5,6,11,17,20,21] For example, flower color from within populations of *T. albanica* is reported to vary from yellow/golden-yellow to carmine-scarlet turning deep reddish maroon,[21] with a range of intermediate colors. Furthermore, the flower color within species may differ in two aspects, first the blotch and the blotch margins may show differences in size and color intensity and second, within some species, anthocyanidins are lacking in certain accessions resulting in yellow or

very light colors.[35] Experiments are based on selection of accessions obtained from natural provenances, as well as mutation experiments with radiation showed that blotch margin and flower color can easily be influenced.[35] Flower color is therefore not regarded as a suitable trait from which to make taxonomic decisions.[2]

Apart from flower morphological features, the characteristics of the bulb tunic have often been used to differentiate between *Tulipa* species and has generally been found to be a reliable character.[36] Our samples of *T. sylvestris* subsp. *sylvestris* and *T. sylvestris* subsp. *australis* both had brownish black tunics, with straight hairs in the inner part of the tunic, located only around the root and on the throat of the bulb. Furthermore, the type and distribution of the trichomes in the tunic of the bulbs of *T. albanica*, *T. kosovarica*, *T. luanica*, and *T. scardica* were also analyzed, here, the trichomes in the form of the straight hairs were located in the inner part of the tunic, densely covering all parts of the tunic. No differences were recorded in the type and distribution of trichomes in the tunics of the bulbs of *T. albanica*, *T. kosovarica*, *T. luanica*, and *T. scardica*.

Species of the *scardica* complex have also been investigated through genome size analyses, providing 2C values for most taxa. Considerable variation has been reported in the 2C value of *T. albanica* with both 54.15 pg[21] and 43.86 pg[22] being reported from separate experiments. *T. kosovarica*, *T. luanica*, and *T. scardica* are recorded as having 45.71 pg, 47.49 pg, and 69 pg 2C values, respectively.[6,22] The incongruent results for *T. albanica* reported (in references 20 and 21) were attributed to the origin of the plant material[22]: leaves collected from wild populations in bloom, vs adult leaves germinated from seeds collected from natural populations. This explanation seems somewhat unconvincing and makes it difficult to base any taxonomic decisions on 2C values for any of these species,

especially given that differences in genome sizes within species could be correlated with differences in habitat,[37] plant phenotype,[38] or caused by technical artifacts.[39] In addition, the DNA content of *T. serbica* has not been measured so this cannot be linked to other species in the *scardica* complex. Overall, this means that our DNA sequence data are likely the best assessment of this species complex to date and should be used as a guide on how to classify these taxa into species over and above current cytogenetic data.

## 4 | MATERIALS AND METHODS

### 4.1 | Plant material

Eight taxa (six species and two subspecies) of the genus *Tulipa* were collected from wild populations between the months of April and May across 2017, 2018, and 2019. All *Tulipa* species were collected in Kosovo, except *T. albanica*, which was collected in Albania (Figure 6). One unidentified plant specimen of *Tulipa* sp. (sample T9, Table 2) was obtained from material provided by the Herbarium of the University Prishtina. *T. kosovarica* (locations Goriç and Koznik) and *T. luanica* (locations Pashtrik and Qafë Prush) were collected from two different localities. Plant specimens were collected, and part of the young leaves was dried in silica gel for DNA extraction. The voucher specimens were deposited at the Herbarium of the University Prishtina, Kosovo and the Emory University Herbarium, Atlanta, USA. Detailed sample information is given in Table 2.

### 4.1.1 | DNA extraction, polymerase chain reaction (PCR), and sequencing

Genomic DNA was extracted from silica gel-dried material or herbarium specimens using the DNeasy Plant Mini Kit (Qiagen Hilden, Germany) according to the manufacturer's instructions. The DNA quality was checked using agarose gel electrophoresis with 1.0% agarose gels containing 0.4 x PeqGreen (VWR, Erlangen, Germany) for 40 minutes at 120 V, which was documented using microDOC system with UV transilluminator (Cleaver Scientific LTD, Rugby, Warwickshire, UK) using 312 nm wavelength.

Extracted DNA was 1:50 diluted with deionized water and then used for PCR. The nuclear internal transcribed spacer region (ITS) and the chloroplast *trnL-trnF*, *psbA-trnH*, and *rbcL* markers were amplified and then sequenced from 23 samples of six species and two subspecies. For a 15-μL PCR reaction, 1 μL of diluted genomic DNA (equivalent to approximately 1-50 ng) was added to 14 μL master mix containing $1 \times$ PCR buffer B, 2.5 mM MgCl$_2$, 130 μM dNTP mix, 0.6 U Taq HOT FIREPol DNA polymerase (all reagents from Solis Biodyne, Tartu, Estonia) and 300 nM forward (ITS5 [5′-GGAAGGAGA AGTCGTAACAAGG-3′;[40] or c [5′-CGAAATCGGTAGACGCTACG-3′;[41] or rbcLaF- ATGTCACCACAAACAGAGACTAAAGC or psbA3′f-GTTATGCATGAACGTAATGCTC) and reverse primers (ITS4 [(5′-TCCTTCCGCTTATTGATATGC-3′;[42] or f [5′-ATTTGAACTGGTGA CACGAG-3′;[41] or rbcL_ajf634R- GAAACGGTCTCTCCAACGCAT or trnHf- CGCGCATGGTGGATTCACAATCC) (Sigma Aldrich, Taufkirchen, Germany). The PCRs were performed in a MIC qPCR

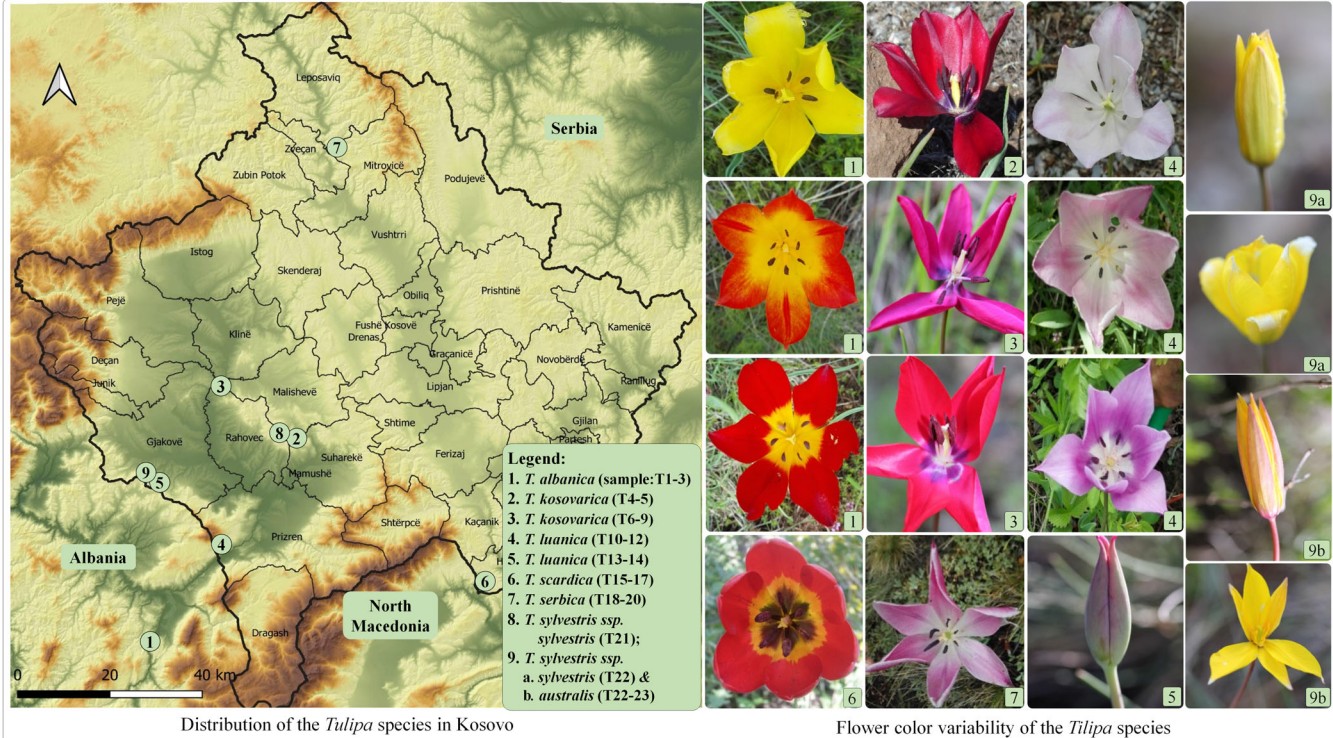

**Distribution of the *Tulipa* species in Kosovo**

**Legend:**
1. *T. albanica* (sample:T1-3)
2. *T. kosovarica* (T4-5)
3. *T. kosovarica* (T6-9)
4. *T. luanica* (T10-12)
5. *T. luanica* (T13-14)
6. *T. scardica* (T15-17)
7. *T. serbica* (T18-20)
8. *T. sylvestris ssp. sylvestris* (T21);
9. *T. sylvestris ssp.*
 a. *sylvestris* (T22) &
 b. *australis* (T22-23)

**Flower color variability of the *Tilipa* species**

**FIGURE 6** Distribution of *Tulipa* sp. in Kosovo and their flower color variability

cycler (Biomloceular Systems, Upper Coomera, Australia). PCR amplifications were performed with an initial denaturation step at 95°C for 14:30 minutes, followed by 40 cycles at 95/58/72°C for 30/30/90 seconds, and a final elongation step of 7 minutes at 72°C. The amplified PCR fragments (2 μL of PCR products) were checked using electrophoresis in 1% agarose gels (low melting point agarose, Sigma Aldrich, Taufkirchen, Germany), using similar conditions as described above for genomic DNA.

Exonuclease I from *Escherichia coli* 20 U/μL (EXO I) and Thermosensitive Alkaline Phosphatase 1 U/μL (FastAP) (Thermo Fisher Scientific Baltics, Vilnius, Lithuania) were premixed in the ratio 1:4 and stored in the freezer. 13 μL PCR products were mixed with 1.3 μL EXO I and FastAP mixture and incubated at 37°C for 15 minutes and 85°C for 15 minutes . Purified PCR products were diluted with distilled water and admixed with sequencing primers according to the requirements of the sequencing company. Sequencing was performed by Microsynth Austria (Vienna, Austria) using Applied Biosystems 3730 × I 96 capillary DNA analyzer (Thermo Fisher Scientific). Every sequence was manually edited with CHROMAS vers. 2.6.6 (Technelysium, South Brisbane, Australia) and aligned with MEGA X software.[43] Edited sequences were subjected to BLAST searches for preliminary analysis.[42]

## 4.2 | Phylogenetic analyses

In total, 106 sequences obtained from 14 taxa were analyzed, 87 of them were newly generated sequences generated from eight *Tulipa* taxa (six species and two subspecies) collected from wild populations in Kosovo and 19 sequences were obtained from GenBank (Table 2). The ITS sequences for *T. ulophylla* (HF952978), *T. tschimganica* (HF952976), *T. sylvestris* subsp. *sylvestris* (HF952974), *T. suaveolens* (MK334468), *T. julia* (HF952964), *T. gesneriana* (MK335217, MK335224), the *trnL-trnF* sequences for *T. ulophylla* (HF953003), *T. tschimganica* (HF953001), *T. sylvestris* subsp. *sylvestris* (HF952999), *T. suaveolens* (HF952998), *T. julia* (HF952989), for rbcL *T. gesneriana* (KP711981), *T. tschimganica* (KM085539), and *T. sylvestri ssp. Sylvestris* (KM085538), were obtained from GenBank. The trees were rooted using *A. edulis* (obtained from GenBank: *ITS* MN173164, *trnL-trnF* HF953006, *rbcL* KC796897, and *psba-trnH* NC034707), *Amana erythronioides* (obtained from GenBank: *ITS* HF952982, *trnL-trnF* HF953007, *rbcL* NC03463, and *psba-trnH* EU939293) and *E. japonicum* (obtained from GenBank: *ITS* EU912083, *trnL-trnF* HF953009, *rbcL* D28156 and *psba-trnH* EU939295) as an outgroup.

ITS, *trnL-trnF*, *psbA-trnH*, and *rbcL* sequences of most of the taxa were amplified and then sequenced from three specimens for each species, while the *T. kosovarica* (locality Goriç) and *T. luanica* (locality Qafë Prush) were amplified and sequenced successfully from two specimens per species. Due to the amplification failure of some specimens (ITS T8 and T10; trnL-trnF T9, T16, and T18), some species were represented by only one or two sequences.

The sequences were aligned using MEGA X software.[43] For ITS analyses, in total 31 sequences were aligned to determine sequence statistics, 21 of them were newly generated, and 10 were obtained from GenBank, for *trnL-trnF* statistical analyses included 28 sequences (20 newly generated and eight obtained from GenBank) (Table 1). For *rbcL* analyses of 29 sequences were used, of them 23 were newly generated and six of them were obtained from Genebank, while for *psbA-trnH* 26 sequences were used for analyses of them 23 newly generated and three obtained from gene bank (outgroup species). Bayesian analyses were conducted through a Markov Chain Monte Carlo (MCMC) approach using BEAST v1.10.4 with the help of BEAGLE v3.1.0 library. The input files for BEAST were prepared in the corresponding BEAUti program and maximum clade credibility trees generated and annotated in TreeAnnotator.[44] The MCMC was run for 10 000 000 generations, with resulting phylogenetic trees sampled every 1000. A burn in period of 1 000 000 was used. All trees were visualized using Figtree (V.1.4.4) and Mega X software.

## 5 | CONCLUSIONS

Our phylogenetic analyses show that Kosovarian tulips can easily be distinguished as either in the subgenera *Eriostemones* or *Tulipa*. Yet, within these subgenera, we found limited resolution to determine clear species relationships using the markers we selected. Nonetheless, we note that there was some genetic distinguishability between the subspecies of *Tulipa sylvestris* (*australis* and *sylvestris*) and that these should therefore continue to be classified as different subspecies but our work does not suggest that they should be raised to species level. In contrast, our data suggest that within the *Tulipa* subgenus, there has been over splitting of species within the *scardica* complex. With our novel genetic perspective, we suggest that *T. luanica* and *T. kosovarica* can be synonymised under *T. serbica*, while both *T. albanica* and *T. scardica* were genetically distinct enough to continue to be treated as species. Further analyses with more extensive sampling and additional genetic markers will be necessary for a better understanding of the natural variability within the taxa of the *scardica* complex, but for now our study provides the most comprehensive genetic understanding of the complex diversity of tulips growing in and around Kosovo. This understanding will not only be crucial for taxonomic stability and future research, but also for identifying conservation priorities, especially given that threats to wild tulips are likely to increase in the near future.[45]

## ACKNOWLEDGEMENTS

We would like to express our thanks to the US Embassy in Kosovo (grant no. US18GR1) for financial support for the sequencing of DNA samples. The author name Xhavit Malaj was corrected to Xhavit Mala after issue publication on September 20, 2021.

## CONFLICT OF INTEREST

We confirm that there are no known conflicts of interest associated with this publication, the manuscript has been read and approved by all named authors and that there are no other persons who satisfied the criteria for authorship but are not listed. We further confirm that the order and contributions of authors listed in the manuscript has been approved by all of us.

## AUTHOR CONTRIBUTIONS

**Avni Hajdari:** Conceptualization, formal analysis, funding acquisition, project administration, resources, supervision, writing-original draft, writing-review, and editing; **Bledar Pulaj**: formal analysis, resources, and visualization; **Corinna Schmiderer**: data curation, writing-review, and editing; **Xahavit Mala**: conceptualization and resources; **Brett Wilson**: writing-review and editing; **Kimete Lluga-Rizani**: investigation and resources; **Behxhet Mustafa**: conceptualization, supervision, writing-review, and editing.

## PEER REVIEW

The peer review history for this article is available at https://publons.com/publon/10.1002/ggn2.202100016.

## ORCID

*Avni Hajdari* https://orcid.org/0000-0001-5688-9679
*Bledar Pulaj* https://orcid.org/0000-0003-0543-090X
*Brett Wilson* https://orcid.org/0000-0001-5181-2875
*Behxhet Mustafa* https://orcid.org/0000-0002-1052-5252

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

## SUPPORTING INFORMATION

Additional supporting information may be found in the online version of the article at the publisher's website.

