## [**Supplementary Information**: Record of Transparent Peer Review · Advanced Genetics]

Record of Transparent Peer Review

A phylogenetic analysis of the wild *Tulipa* species (Liliaceae) of Kosovo based on plastid and nuclear DNA sequence

Avni Hajdari*, Bledar Pulaj, Corinna Schmiderer, Xhavit Malaj, Brett Wilson, Kimete Lluga-Rizani, Behxhet Mustafa

*Corresponding

Review timeline:	Date Submitted: 28-Jul-2020
	1 st Peer Review 03-Aug-2020 to 13-Aug-2020
	Editorial Decision: 13-Aug-2020 Reject, invite appeal to permit revision
	Revision Received: 12-May-2021
	2 nd Peer Review 17-May-2021 to 29-Jun-2021
	Editorial Decision 29-May-2021 Minor Revision
	Revision Received: 26-Jul-2021
	Editorial Decision: 27-Jul-2021 Accept in Principle
	Accepted: 02-Aug-2021

Editor: Myles Axton

Initial Editorial Evaluation	30-Jul-2020
-------------

Scope

Do the research, methods or topics fit within the aims of this, or another journal?

-This journal would be very interested in population genetics, hybridization and incipient speciation of Kosovan *Tulipa* studied by genome wide SNPs (genotyping by sequencing), STRs and whole plastid sequence. We are fascinated by the history of tulips as people moved from Central Asian Steppe to Middle East and Europe. This article aims to set a foundation for understanding tulip radiation.

Conceptual advance

What is already known in this area and related fields?

Turktas et al. 2013 (cited) barcoded and compared Turkish tulips, identified three of the subgenus clades but using only trnL-trnF and ITS probably had too few characters to identify the Clusianae group. They used *Lilium sulphureum* outgroup.

Christenhusz et al. 2013 DOI: 10.1111/boj.12061 studied 25 *Tulipa* taxa (of the 76 morphological taxa known). They sequenced five plastid regions (trnL intron and trnL-trnF spacer, rpl16 intron, rps12-rpl20 intergenic spacer and matK) and the internal transcribed spacer (ITS) region of nuclear ribosomal DNA. 2 *Amana* and 2 *Erythronium* species were used as outgroups. Visually aligned sequences in PAUP and coded only 6 indel characters. MP supported monophyletic *Tulipa* with four subclades, Bayesian analysis also strongly supported a single genus with the same four subgenera.

Kritskaya, et al. Genetic diversity of *Tulipa suaveolens* (Liliaceae) and its evolutionary relationship with early cultivars of *T. gesneriana* Plant Syst. Evol. 306 (2), 33 (2020) DOI: 10.1007/s00606-020-01667-7. Barcoded psbE-petL and ITS.

What gap in knowledge motivates this research?

-Understanding the morphological and growth adaptations of wild tulip radiation

How do the main claims of this study relate to benchmark prior publications?

-This uses the tools of the field to extend the work of Turktas et al. 2013, Christenhusz et al. 2013 and Kritskaya, et al.

1 st Peer Review	03-Aug-2020 to 13-Aug-2020
----------------------------

Reviewer #1

This manuscript at present is immature for formally published because the following questions. I don't think the authors have clearly expressed the scientific significance of this study in the introduction part.

1.1 Logically, the objects of phylogenetic study should be natural units. So, it is necessary to state clearly that the members of Kosovo *Tulipa* are the natural units or not.

1.2 Additionally, the weakly support for the phylogenetic relationships of the manuscript cannot provide evidence for classification according to the abstract and conclusion. Molecular results should be an important evidence for classification, but is not the only evidence, especially the author calculated only two sequences.

Reviewer #2

This paper suffers of several flaws:

- 2.1-the accuracy of identification is not adequate. The authors also took sequences from genbank, but this should be done after having verified the connected specimens. There is a "Tulipa sp.", a "Tulipa sylvestris" [without] subspecies indication etc.
- 2.2-the two markers used are evidently not adequate to resolve a phylogeny of these Tulipa taxa. More variable markers should be added to the analysis, which is otherwise somehow unuseful, distinguishing only the two main groups (subgenera).
- 2.3-the root used for the trees is not adequate. At least another representative of a different genus of Tulipeae should be added.
- 2.4-it is completely unuseful to carry out three analyses, without justification, just because the software used (PAUP) is able to do them! Moreover, a Bayesian approach would have been much more in line with current standards.
- 2.5-the quality of ITS sequences (looking for pseudogenes) should be checked before their inclusion in the analyses.
- 2.6-some relevant recent Literature is missing and also the discussion on Tulipa sylvestris and T. pumila is completely missing. I also attach an annotated version of the manuscript with more details and comments.

Comments on the manuscript

- 2s.1 Placement of Tulipa in Liliaceae: <https://www.tandfonline.com/doi/abs/10.1080/11263504.2015.1115435>
- 2s.2 [T. sylvestris "subspecies"] are sufficiently different to be treated as different species. See for instance: <https://esj-journals.onlinelibrary.wiley.com/doi/abs/10.1111/1442-1984.12267>
<https://www.cambridge.org/core/journals/seed-science-research/article/two-closely-related-tulipa-species-with-different-ploidy-levels-show-distinct-germination-rates/>
- 2s.3 Barcoding and phylogenetic analysis are different procedures
- 2s.4 Lilium belongs to Liliaceae, adding some other representative of Tulipeae genera would be much more desirable

Reviewer #3

- 3.0 The article is devoted to the study of genetic diversity and taxonomic revision of natural tulips growing in Kosovo. The relevance of the study is beyond doubt, since the taxa presented are really poorly understood. In general, the manuscript is well written and understandably, however, there are some comments:
- 3.1 the article will be perceived much better if figure or electronic supplementary material is included with photographs of the studied taxa;
- 3.2 I have not found information about data deposition in the manuscript. It is desirable that the data be loaded into the repository (TreeBase or Dryad);
- 3.3 This article uses sequences of other tulip species downloaded from GenBank. It will not be superfluous to add links to related articles. The sequences were taken from two sources - Christenhusz et al. (2013) and Kritskaya et al. (2020). There is a link to the first of these articles, but for some reason there is no link to the second;
- 3.4 in table 1 GenBank accession number for Tulipa suaveolens is given with an error;
- 3.5 the text contains Latin names in which the specific epithets begin with a capital letter.

1 st Editorial Decision	13-Aug-2020
-------------

Editorial decision: Reviewers #1 and #2 do not regard the current marker selection has sufficient variation to demonstrate whether Balkan tulips of the *T. scardica* group are or are not monophyletic or consistent with separate species level taxa. Reviewer #3 recognises the value of this contribution since the genetic variation and relationships among these tulips remains unsolved. We have decided to **reject** the manuscript in its current form. But, we are keen to reconsider the work should the authors be able to provide the requested documentation and analysis of molecular and phenotypic markers sufficient to test the three hypotheses suggested below (ED1). **We will consider an appeal** of this decision once the authors experimentally address the specific recommendations of all of the reviewers and provide a point-by-point reply to all of our comments.

Editor's understanding of the reviews

- Reviewer #1 Advocates rejection in current form
Reviewer #2 Advocates rejection in current form
Reviewer #3 Advocates minor revision

Reviewer comments	Editor recommendation	Author reply	Changes to Manuscript
1.1 it is necessary to state clearly that the members of Kosovo Tulipa are the natural units or not. 2s.2 [T. sylvestris “subspecies”] are sufficiently different to be treated as different species 3.0 The relevance of the study is beyond doubt, since the taxa presented are really poorly understood.	ED1 The evidence for Kosovo Tulipa as separate species (and also any evidence for other Balkan Tulipa) should be discussed in the light of the hypothesis that T. scardica type tulips are present in many forms. This would lead to three clear hypotheses:  1) Biologically distinct species 2) One interbreeding species with population variation 3) Hybridising complex with or without survival of progenitor species Testing these hypotheses could be done by literature and herbarium review of morphological characters, together with appropriate selection of genome-wide SNPs or microsatellite variants	1.1. It clearly state that the members of Kosovo Tulipa are the natural units in Kosovo. 2s.2 the taxonomic position of the “subspecies” of T. sylvestris is clarified 3.0 Further research were carried to clarify, if the analyzed taxa are biologically distinct species. Entire manuscript were revised in this regard, some main finding are indicated in next column.	All changes are indicated in the manuscript with track-changes. Introduction: pp 3. In Kosovo, the genus Tulipa is represented by eight taxa (six species and two subspecies), belonging to the subgenera Eriostemones and Tulipa. In general, researchers working on these species have used different morphological traits to define the taxonomic relationship between them. Discussion: pp 10. Among the newly sequenced species of the Eriostemones clade, there was little distinguishable difference between Tulipa sylvestris subsp. australis (Link) Pamp and Tulipa sylvestris subsp. sylvestris from Kosovo. Our work therefore suggests that these subspecies should continue to be classified under the species Tulipa sylvestris. Conclusions: pp 16, Yet, limited distinguishability between specimens under the species name Tulipa sylvestris was found providing some support that the subspecies australis and sylvestris should not be raised to species level. Discussion: pp 10, Our work therefore suggests that these subspecies should continue to be classified under the species Tulipa sylvestris. In the subgenus Tulipa, the grouping of the species T. scardica, T. serbica, T. albanica, T. kosovarica and T. luanica in

			one clade provides strong evidence of a close relationship between these taxa, confirming the existence of the scardica complex distributed across the southern Balkans (Christenhusz et al., 2013). Our analysis highlighted the genetic distinctness of T. albanica and T. scardica from the other species in this complex. This provides evidence for the over splitting of this complex and the need to synonymise some of the taxa under one species name, specifically T. luanica and T. kosovarica under T. serbica. Conclusions: pp 16, Whilst our data suggests that within the Tulipa subgenus there has been over splitting of species within the scardica complex. With our novel genetic perspective, we suggest that Tulipa luanica and Tulipa kosovarica can be synonymised under Tulipa serbica, whilst both Tulipa albanica and Tulipa scardica were genetically distinct enough to continue to be treated as species.
1.2 the weak support for the phylogenetic relationships of the manuscript cannot provide evidence for classification according to the abstract and conclusion. Molecular results should be an important evidence for classification, but is not the only evidence, especially the author calculated only two sequences. 2.2 the two markers used are evidently not adequate to resolve a phylogeny of these Tulipa taxa. More variable markers should be added to the analysis, which is otherwise somehow	ED2 Markers appropriate to the level of taxonomic discrimination needed to test explicit hypotheses should be employed since the genomic regions used lack sufficient variation to discriminate below the subgenus level	Two additional markers (rbcL and psbA-trnH), as well as a combined dataset of all used markers employed to get an appropriate level of taxonomic discrimination needed. The discrimination level increases by using markers, especially by a combination of all markers in the dataset.	The introduction, method, results, discussion and conclusions section were rearranged accordingly. It is indicated in the manuscript with track-changes.

unuseful, distinguishing only the two main groups (subgenera).			
2.3-the root used for the trees is not adequate. 2s.4 Lilium belongs to Lillieae, adding some other representative of Tulipeae genera would be much more desirable	ED3 Outgroup within Tulipeae would be more appropriate than L. martagon	Instead of L. martagon , Amana edulis was used as an outgroup species.	It is indicated in the manuscript with track-changes. Pp 15 and pp 22.
2.6-some relevant recent Literature is missing and also the discussion on Tulipa sylvestris and T. pumila is completely missing. I also attach an annotated version of the manuscript with more details and comments. 3.3 The sequences were taken from two sources - Christenhusz et al. (2013) and Kritskaya et al. (2020). There is a link to the first of these articles, but for some reason there is no link to the second;	ED4 Add and discuss the relevant references	Additional literature added in the discussion section.	They are indicated in the manuscript with track-changes. - Bruni et al. 2010. - Hayashi et al. 2000. - Kress and Erickson 2007. - Kritskaya et al. 2020. - Lahaye et al. 2008. - Peruzzi et al. 2016. - Ronquist et al. 2012. - Wilson et al. 2021.
1.2 ...Molecular results should be an important evidence for classification, but is not the only evidence.. 2.1-the accuracy of identification is not adequate. The authors also took sequences from genbank, but this should be done after having verified the connected specimens. There is a " Tulipa sp. ", a " Tulipa sylvestris " [without] subspecies indication etc. 3.1 the article will be perceived much better if figure or electronic supplementary material is included with photographs of the studied taxa;	ED5 Link Genbank sequences to their vouchers and provide photographic documentation of the type specimens of all sequenced taxa and phenotypic variation described	Additionally, figure 6 provided. The photographic documentation of the newly analyzed sequences.	It is indicated in the manuscript with track-changes, pp 28.
2.4-it is completely unuseful to carry out three analyses, without justification, just because the software used (PAUP) is able to do them! Moreover, a Bayesian approach would have been much more in line with current standards.	ED6 Avoid loss of statistical power from testing multiple hypotheses. Do only the experiments required to test the hypothesised taxonomic relationships. Consider using sequence variation to provide Bayesian support for alternative taxonomic hypotheses.	Bayesian interference approach was used to construct phylogenetic trees, using MrBayes software.	All phylogenetic tree are replaced with those generate by MrBayes, pp23-28. Method, result, discussion and conclusion section are revised based on new findings.

Reviewer #4

Despite some of the shortcomings I have mentioned in the article below, researchers should be encouraged in this study aimed at solving this species complex in the Balkans. As far as I understand, the article was developed in line with the suggestions of 3 referees before. At this point, I have a few suggestions that will raise the level of the article in terms of morphological and molecular:

4.1 In the morphological evaluations, the tunic hair type and their placement are quite distinctive characters at the beginning of the possible ancestral and most important characters for the genus *Tulipa*. Researchers generally used flower features in morphological comparisons. However, the characteristics of tunician feathers will reveal the similarities and differences to a great extent, and with a few exceptions, it is generally a reliable character (See Eker 2019).

Eker İ (2019). Importance of Tunic Hair Settlement and Flower Colour Variation Range in Taxonomy of The Genus *Tulipa* L. *Bağbahçe Bilim Dergisi* 6(2): 1-9.

4.2 In the article, the authors state that there is no significant difference between *Tulipa sylvestris* subspecies. However, there are two views that the probable gene center of the ancestral diploid subspecies (subsp. *australis*) is either in western Anatolia or in North Africa. The populations of this taxon spreading in Europe are mostly polyploid (triploid, tetraploid) subspecies (subsp. *sylvestris*) that were later migrated and naturalized, and it has been revealed that there are important morphological, chorological and cytotoxic differences between them (See Eker et al. 2014). Researchers should be sure about the subspecies of the taxa they use in their analysis. I have a concern that all of the populations used may be subsp. *sylvestris*. Otherwise, this may lead to wrong evaluations. The shortest way to solve this is to look at the chromosome numbers of the populations if there is not enough morphological or molecular evidence.

Eker İ, Babaç MT, Koyuncu M (2014). Revision of the genus *Tulipa* L. (Liliaceae) in Turkey. *Phytotaxa* 157(1): 001-112.

4.3 The phylogeny inference based on *psbA-trnH*, *trnL-F* regions distinguished two major groups, belonging to the subgenera *Eriostemones* and *Tulipa*. In my opinion, the data set in the article can be increased by adding new markers and phylogenetic analysis can be developed to reveal better resolved trees.

4.4 The outgroup taxa can be increased (at least 3 taxa) in order to get better resolutions. For Markov Chain Monte Carlo (MCMC), 1 million generations seems not to be enough and can be increased. The authors may use BEAST instead of MrBayes. After these modifications, if authors still get polytomy then they would better switch to NGS methods instead of Sanger sequencing.

Reviewer #5

I read carefully the reviewed manuscript:1720008/j50sWSi4. No further comments from me. The manuscript can be accepted

Editorial decision: Minor Revision

Incorporate Reviewer 4's excellent constructive criticisms and revise the manuscript according to the four editorial recommendation

Editor's understanding of the reviews

Reviewer #4 Recommends Major Revision

Reviewer #5 Recommends Accept

Reviewer comments	Editor recommendation	Author reply	Changes to Manuscript
4.1 Researchers generally used flower features in morphological comparisons. However, the characteristics of tunician feathers will reveal the similarities and differences to a great extent, and with a few exceptions, it is generally a reliable character (See Eker 2019).	ED1 Cite this reference and commend on the distribution of the tunic hair character in the taxa examined.	The reference is cited and the distribution of the tunic hair character in the taxa examined is analyzed.	Apart from flower morphological features, the characteristics of the bulb tunic have often been used to differentiate between Tulipa species and has generally been found to be a reliable character (Eker, 2019). Our samples of T. sylvestris subsp. sylvestris and T. sylvestris subsp. australis both had brownish black tunics, with straight hairs in the inner part of the tunic, located only around the root and on the throat of the bulb. Furthermore, the type and distribution of the trichomes in the tunic of the bulbs of T. albanica, T. kosovarica, T. luanica and T. scardica were also analysed, here the trichomes in the form of the straight hairs were located in the inner part of the tunic, densely covering all parts of the tunic. No differences were recorded in the type and distribution of trichomes in the tunics of the bulbs of T. albanica, T. kosovarica, T. luanica and T. scardica.
4.2 I have a concern that all of the populations used may be subsp. sylvestris. Otherwise, this may lead to wrong evaluations. The shortest way to solve this is to look at the chromosome numbers of the populations if there is not enough morphological or molecular evidence.	ED2 Check morphological characters and if necessary chromosome number to ensure that all populations used fit the subspecies sylvestris as described by Eker 2014. Discuss and cite this reference.	The characteristics of tunica are checked, actually we could not check the chromosome number of the species, but is indicated as limitation. A short discussion added in this regard and the references is cited.	These subspecies are known to have differing chromosome numbers, with T. sylvestris subsp. australis a diploid form of Tulipa sylvestris, and Tulipa sylvestris subsp. sylvestris encompassing triploid or tetraploid forms of Tulipa sylvestris (Eker et al., 2014). Yet, the native range of these subspecies remains unclear, and many morphologically intermediate forms are known to occur in the wild (Christenhusz et al., 2013).

			Further cytotaxonomic studies will therefore be needed to investigate the chromosome numbers of the specimens located in Kosovo to confirm their taxonomic identity, whilst extensive in-depth molecular work will be needed to unentangle this widespread, complicated taxon.
4.3 The phylogeny inference based on psbA-trnH, trnL-F regions distinguished two major groups, belonging to the subgenera Eriostemones and Tulipa . In my opinion, the data set in the article can be increased by adding new markers and phylogenetic analysis can be developed to reveal better resolved trees.	ED3 If you have data for more markers, please offer the more resolved tree. Otherwise, discuss the limitations of this dataset for this particular analysis.	Actually we do not have data for more markers. It is indicated as limitation in manuscript.	There is significant limitations in our assessment of sections of the genus Tulipa both in terms of the genetic marker used as well as in the extremely poor species representation. We therefore do not make any conclusive statements about the use of sections in the genus Tulipa but do note that these may not all hold as more genetic data becomes available.
4.4 The outgroup taxa can be increased (at least 3 taxa) in order to get better resolutions. For Markov Chain Monte Carlo (MCMC), 1 million generations seems not to be enough and can be increased. The authors may use BEAST instead of MrBayes. After these modifications, if authors still get polytomy then they would better switch to NGS methods instead of Sanger sequencing.	ED4 Add three or more outgroup taxa and repeat MCMC with an appropriate number of generations. Use BEAST and comment on the results. If there are still too few markers for resolution, discuss how many might be needed and the limitations of the current sequence data.	Three outgroup taxa are used: Amana edulis , Amana erythronioides and Erythronium japonicum , respectively. The BEAST is used to construct the phylogenetic trees, and results are commented in the light of the new information provided from phylogenetic analyses. The changes are indicated directly with track change in manuscript.	The trees were rooted using Amana edulis (obtained from GenBank: ITS MN173164, trnL-trnF HF953006, rbcL KC796897 and psbA-trnH NC034707), Amana erythronioides (obtained from GenBank: ITS HF952982, trnL-trnF HF953007, rbcL NC03463 and psbA-trnH EU939293) and Erythronium japonicum (obtained from GenBank: ITS EU912083, trnL-trnF HF953009, rbcL D28156 and psbA-trnH EU939295) as an outgroup. Limitation regarding the used markers are indicated in manuscript as it is mention above.

2nd Editorial Decision 27-Jul-2021

Manuscript has been suitably revised in light of the remaining reviewer's recommendations and my editorial suggestions. The manuscript is now accepted in principle subject to minor formatting edits to suit the journal's style.

3rd Editorial Decision 02-Aug-2021

The article is accepted for publication.